# Pulsed vector atomic magnetometer using an alternating fast-rotating field

Tao Wang [1,2] ✉, Wonjae Lee [1,4], Mark Limes[3,5], Thomas Kornack[3], Elizabeth Foley[3] & Michael Romalis[1] ✉

We introduce a vector atomic magnetometer that employs a fast-rotating magnetic field applied to a pulsed $^{87}$Rb scalar atomic magnetometer. This approach enables simultaneous measurements of the total magnetic field and its two polar angles relative to the rotation plane. Operating in gradiometer mode, the magnetometer achieves a total field gradient sensitivity of 35 fT/$\sqrt{\text{Hz}}$ (0.7 parts per billion) and angular resolutions of 6 nrad/$\sqrt{\text{Hz}}$ at a 50 $\mu$T Earth field strength. The noise spectra remain flat down to 1 Hz and 0.1 Hz, respectively. Here we show that this method overcomes several metrological challenges commonly faced by vector magnetometers and gradiometers. We propose a unique peak-altering modulation technique to mitigate systematic effects, including a newly identified dynamic heading error. Additionally, we establish the fundamental sensitivity limits of the sensor, demonstrating that its vector sensitivity approaches scalar sensitivity while preserving the inherent accuracy and calibration benefits of scalar sensors. This high-dynamic-range, ultrahigh-resolution magnetometer offers exceptional versatility for diverse applications.

The measurement of magnetic fields has a long and rich history, beginning with the invention of the compass. The earliest known compass, the "south-governor" (sinan), dates back over 2000 years. Among the first and still, most widely used vector sensors are fluxgate magnetometers based on asymmetry in saturation of a soft magnetic material in an applied oscillating magnetic field[1]. Advanced condensed-matter-based approaches involve SQUID magnetometers[2], Hall sensors[3] and magneto-resistive sensors[4]. These sensors are intrinsically vector devices that measure one component of the magnetic field, so three separate sensors are typically required for full vector sensing. For example, the ESA Swarm mission necessitates three-axis fluxgate magnetometers and an Absolute Scalar Magnetometer for calibration[5]. In contrast, atomic magnetometers are inherently scalar devices that gauge the energy gap between spin states in a magnetic field, detecting a frequency directly proportional to the field strength. This frequency-based measurement approach allows scalar magnetometers to achieve very high relative precision. Additional interactions are required to define the vector axes for a scalar sensor.

Sensitive vector magnetometers operating in Earth's field have several metrological challenges. For example, a 50 fT magnetic field corresponds to 1 part in $10^9$ of Earth's field. Such relative precision is beyond the capability of most analog voltage measurements. However, frequency measurements can easily achieve much higher relative precision. SQUID magnetometers also can achieve higher fractional resolution using fluxquanta counting techniques[2]. Another challenge is a stable alignment and orthogonality of vector axes directions, which require nano-radian stability. We use an applied rotating field to create vector sensitivity. For any coil imperfections or non-orthogonality, as long as the coils remain linear, a rotating field always has a unique rotation plane that defines the vector axes in our approach.

Atomic vector sensors operating in the spin-exchange relaxation-free (SERF) regime are considered the most sensitive[6]; however, their

[1]Department of Physics, Princeton University, Princeton, NJ, USA. [2]A*STAR Quantum Innovation Centre (Q.InC), Institute of Materials Research and Engineering (IMRE), Agency for Science, Technology and Research (A*STAR), 2 Fusionopolis Way, 08-03, Singapore, Republic of Singapore. [3]Twinleaf LLC, 300 Deer Creek Dr., Plainsboro, New Jersey, USA. [4]Present address: Department of Physics, Harvard University, Cambridge, MA, USA. [5]Present address: Virginia Tech, Blacksburg, Virginia, USA. ✉e-mail: tao_wang@imre.a-star.edu.sg; romalis@princeton.edu

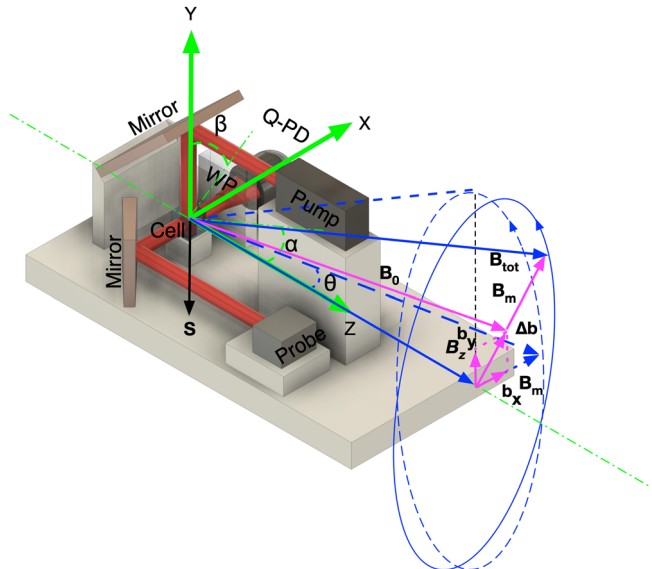

**Fig. 1 | Experimental setup: Three coils determine the x, y, and z directions.** The sensor head sits on a rotating stage with the cell positioned at the stage's center. The sensor head can freely rotate in the xz and yz-planes. $\alpha$ is the angle by which the probe beam moves away from the position where the probe beam is perpendicular to the static magnetic field in xz-plane. $\beta$ is the angle by which the pump beam moves away from the position where the pump beam is perpendicular to the static magnetic field in yz-plane. $\theta$ represents angle between the axis of rotation and the rotating field vector. Notably, WP stands for Wollaston prism, and Q-PD stands for quadrant photodiode.

low dynamic range limits their applications. Scalar magnetometers based on multi-pass cells have also reached the sub-fT/$\sqrt{\text{Hz}}$ sensitivity regime[7]. Scalar atomic magnetometers have been widely used outside the lab in various geomagnetic and space applications[8]. Recently, they have also been used to detect biomagnetic fields outside the lab[9]. Vector sensing of similar sensitivity will provide better localization accuracy for detecting magnetic sources[10], whether in the Earth or the brain.

Several different approaches have created vector sensitivity for atomic sensors. For instance, variometers based on scalar magnetometers[11–14], magnetic field modulation[15,16], multi-laser beams to measure multi-spin projections[17–20], nonlinear magneto-optical rotation (NMOR) magnetometers based on atomic alignment[21,22], and vector magnetometry based on magneto-optical phenomena[23,24]. In particular, all-optical vector magnetometers can be achieved by modulation of the light shift, which acts as a pseudo-magnetic field modulation[25]. Lastly, extracting magnetic field vector information by using microwaves as a 3D spatial reference[26]. While all of these methods measure the amplitudes or frequencies of Larmor precession, frequency measurements can achieve higher fractional precision and are less affected by drifts in cell temperature, pump and probe laser powers, and wavelengths. We focus on measuring the zero-crossings of the Larmor precession signals, which not only provide frequency information but also capture phase shifts and integrate magnetic field modulations, offering more comprehensive insights into the magnetic fields.

We introduce a Fast Rotating Field (FRF) vector magnetometer that can extract three-axis vector information without compromising its scalar resolution. This is achieved by adding a fast rotating magnetic field−at a rotation rate exceeding the spin relaxation rate−to compact scalar magnetometers. Utilizing multipass cells, these scalar magnetometers achieve sensitivities as low as several fT/$\sqrt{\text{Hz}}$ in the earth's ambient environment[9,27].

Our method involves a scalar magnetometer configuration using a pulsed pump laser similar to[9], along with the application of a fast-

rotating magnetic field. This entire sensor can be miniaturized to a very compact size using microelectromechanical systems techniques[28–31].

We conducted a thorough study on the systematic effects of the magnetometer and developed a unique modulation technique called peak-altering fast rotating field modulation to cancel out these systematic errors. The magnetometer's performance is evaluated, and we propose analytical fundamental limits for the sensitivity of the vector free-spin-precession magnetometer based on the Cramer-Rao Lower Bound (CRLB).

## Results

### Spin dynamics

The complete description of spin evolution involves density matrix theory[32]. When ground states are distributed according to spin temperature, spin evolution can be effectively described by the Bloch equations[33,34].

$$\frac{d\mathbf{P}}{dt} = \gamma \mathbf{P} \times \mathbf{B}, \tag{1}$$

where $\gamma$ represents the gyromagnetic ratio, $\mathbf{P}$ is spin polarization, and $\mathbf{B}$ is magnetic field. The approximation of rotation matrices is utilized to assess spin evolution in the rotating frame, enabling the derivation of analytical solutions to the time-dependent Bloch equation.

As shown in Fig. 1, we assume the spins are fully polarized along the negative y-axis. The magnetic field initially points along the z-axis, while a rotating field originates from the x-axis and rotates counterclockwise in the xy-plane. We have $B_x = B_m \sin(\omega_m t + \phi_x)$, $B_y = B_m \sin(\omega_m t + \phi_y)$, and $|\phi_x - \phi_y| = \pi/2$. For this assumption, $\phi_x = \pi/2$ and $\phi_y = 0$. We consider the behavior of polarization in a magnetic field, where it undergoes precession in response to the applied field. This magnetic field is composed of both a static and a rotating component. In the presence of these components, the polarization vector evolves according to specific rotations. The spin evolution, without considering spin relaxations, can be expressed as:

$$\mathbf{P}(t) = \mathcal{R}[\theta, \hat{z}] \cdot \mathcal{R}[\psi, \tilde{\mathbf{B}}_{tot}] \cdot (-\hat{y}), \tag{2}$$

where $\mathcal{R}[\phi, \mathbf{v}]$ is a 3D rotation matrix for an anti-clockwise rotation of $\phi$ degrees around the vector $\mathbf{v}$. In the co-rotating reference frame, the effective field after applying the rotating wave approximation is given by $\tilde{\mathbf{B}}_{tot} = (B_m, 0, B_z - \omega_m/\gamma)$. Here, $\hat{x}, \hat{y}$ and $\hat{z}$ are the unit vectors along the x, y and z axes, respectively. We define $\theta = \omega_m t$ and $\psi = \gamma t |\tilde{\mathbf{B}}_{tot}|$. Eventually, the spin projections can be written as:

$$P_x(t) = \hat{x} \cdot \mathbf{P}(t) = \cos \omega_0 t \sin \omega_m t + \frac{\gamma B_z - \omega_m}{\omega_0} \sin \omega_0 t \cos \omega_m t, \tag{3}$$

$$P_z(t) = \hat{z} \cdot \mathbf{P}(t) = -\frac{\gamma B_m}{\omega_0} \sin \omega_0 t, \tag{4}$$

where $\omega_0 = \gamma |\tilde{\mathbf{B}}_{tot}|$. By inserting residual transverse magnetic fields $b_x$ and $b_y$ into Eq. (2), we can get

$$
\begin{aligned}
P_x(t) \approx & \cos\left(\omega_0 t + \gamma \frac{B_m b_y \cos \omega_m t - B_m b_x \sin \omega_m t}{\omega_m \sqrt{B_m^2 + B_z^2}}\right) \sin \omega_m t \\
& + \sin\left(\omega_0 t + \gamma \frac{B_m b_y \cos \omega_m t - B_m b_x \sin \omega_m t}{\omega_m \sqrt{B_m^2 + B_z^2}}\right) \cos \omega_m t \cdot \frac{\gamma B_z - \omega_m}{\omega_0},
\end{aligned} \tag{5}
$$

where $|\tilde{\mathbf{B}}_{tot}| = \sqrt{B_m^2 + b_x^2 + b_y^2 + (B_z - \omega_m/\gamma)^2}$. The residual magnetic fields in the x-direction, $b_x$, and y-direction, $b_y$, are proportional to the amplitudes of the first harmonic, $\sin \omega_m t$ and $\cos \omega_m t$, respectively, in

**Table 1 | The signs of the systematics as a function of the phases of the rotating magnetic field**

| Shot No. | $\phi_x$ | $\phi_y$ | $b_x$ | $b_y$ | $\phi_B$ | $\phi_{prob}$ | $B_{PD}$ | $\phi_{2nd}$ |
|---|---|---|---|---|---|---|---|---|
| 1 | $\pi/2$ | 0 | – | + | + | – | + | – |
| 2 | $\pi/2$ | $\pi$ | – | – | – | + | + | + |
| 3 | $3\pi/2$ | $\pi$ | + | – | + | – | – | – |
| 4 | $3\pi/2$ | $2\pi$ | + | + | – | + | – | + |

the phase shift. The conversion factor for converting from the amplitude of the phase shift to the transverse magnetic field is given by $\pm \omega_m \sqrt{B_m^2 + B_z^2}/\gamma B_m$ (Table 1 details the sign dependence for $b_x$ and $b_y$ across various starting phases). Thus, by analyzing the phase shifts of the Larmor precession signal under the fast rotating field, we can measure both the transverse magnetic fields and the total magnetic field.

The intuitive classical model underlying the operating principle of the FRF vector magnetometer is illustrated in Fig. 1. The scalar magnitude of the total magnetic field, $|\mathbf{B_{tot}}|$, depends on the angle between the rotating field $\mathbf{B_m}$ and the residual magnetic field $\mathbf{B_0}$. When a small residual transverse magnetic field $\Delta\mathbf{b}$—comprising $\mathbf{b_x}$ and $\mathbf{b_y}$—is present, $|\mathbf{B_{tot}}|$ varies with oscillating components at the rotation frequency of $\mathbf{B_m}$. The phases and amplitudes of these oscillations are influenced by the angle between $\mathbf{B_0}$ and $\mathbf{B_m}$. Consequently, the Larmor precession frequency, proportional to $|\mathbf{B_{tot}}|$, also displays oscillating components: a residual field in the x-direction, $b_x$, produces an out-of-phase oscillation, while a residual field in the y-direction induces an in-phase component.

This model provides a fundamental description of the magnetometer's operation. However, for precise measurements, several systematic effects—some even resulting from hyperfine interactions—require in-depth analysis to account for and minimize potential measurement biases.

## Systematics

To achieve high accuracy, systematic errors must be carefully studied and eliminated. In the FRF vector magnetometer, several systematic effects are present. By expanding Eq. (3) into a power series and examining the phase shifts to second order in $B_m$, we can identify two phase-shift contributions from $B_m$:

**Berry's phase shift.** The Berry phase shift refers to the geometric phase acquired by a spin system during an adiabatic, cyclic evolution[35]. As the magnetic field rotates slowly, the spins adiabatically follow the field. At the end of a complete cycle, when the field returns to its original configuration, the spins acquire a phase shift. If the system undergoes cyclic adiabatic evolution at a constant frequency, the Berry phase accumulates linearly over time. Thus, the Berry phase term as a function of time, from Eq. (3), can be written as

$$\phi_B(t) = \frac{B_m^2 \omega_m t}{2B_z^2} \approx (1 - \cos\theta)\omega_m t, \qquad (6)$$

where $\theta \approx B_m/B_z$ is the angle between the total magnetic field and z-axis as shown in Fig. 1. The Berry's phase acquired over one complete period, given by Eq. (6) as $2\pi(1 - \cos\theta)$, is consistent with Berry's phase shift of Nuclear Magnetic Resonance (NMR) described in Ref. 35.

**Second harmonic phase shift.** Another phase shift from Eq. (3) is proportional to $\sin(2\omega_m t)$, which can be written as

$$\phi_{2nd}(t) \approx -\frac{B_m^2}{4B_z^2} \sin(2\omega_m t). \qquad (7)$$

This phenomenon leads to a non-zero second harmonic phase shift, even in the absence of a residual magnetic field and when the rotating field is perfectly symmetric in the xy-plane. Any asymmetry in the rotating field along the x and y-directions also produces a second harmonic signal instead of the first harmonic at $\omega_m$, ensuring that it does not affect the measurement of transverse magnetic fields.

**Heading errors.** The heading error of atomic magnetometers refers to the dependence of the measured magnetic field values on the orientation of the sensor relative to the magnetic field[36]. The pump beam heading error is well-studied and is primarily caused by nonlinear Zeeman splitting and the difference between Zeeman resonances in the two hyperfine ground states. We set nuclear Landé factor $g_I \approx 0$[37] and the heading error can be well described by $B_{SH} = \mathcal{B}_\mathcal{H} \sin\beta$, where

$$\mathcal{B}_\mathcal{H} \approx \frac{P(7+P^2)}{5+3P^2} \frac{3h\gamma B_{tot}^2}{4\pi A_{hf}}, \qquad (8)$$

$\beta$ is the angle by which the pump beam moves away from the position where the pump laser is perpendicular to the magnetic field. $A_{hf} = h \cdot 3.417$ GHz is the hyperfine structure constant for ground state[38], $h$ is Planck constant. We refer it to as the static heading error (to distinguish it from the dynamic heading error introduced below, we refer to this conventional heading error as static heading error), which depends on the sensor's orientation with respect to the static magnetic field. We introduce the concept of the dynamic heading error, which occurs when the total magnetic field is rotating rather than static.

We discovered that the dynamic heading error can be explained by the fact that the spin precession plane adiabatically follows the rotation of the total magnetic field. This maintains a constant relative angle between the total magnetic field vector and the spin precession plane, resulting in a stable precession frequency, as the weighted sum of different Zeeman transitions remains consistent. Consequently, the dynamic heading error depends on the initial angle between the spin polarization and the total magnetic field, which remains constant and equal to its initial value. (More details in Sec. 2.3).

To clarify the elimination of the systematic dynamic heading error, we separate the dynamic heading error, $B_{DH}$, into two components: a static part ($B_{SH} = \mathcal{B}_\mathcal{H} \cos\theta \sin\beta$), which is equivalent to the conventional heading error in the absence of the rotating field ($\approx \mathcal{B}_\mathcal{H} \sin\beta$) when $\theta$ is small, and a rotating-field-phase-dependent part ($B_{PD}$). Notably, the static heading error is independent of the rotating field phases.

There's an analogous "heading error" effect from the probe beam. This error creates a systematic effect where the measured transverse magnetic field depends on the relative angle between the probe beam and the plane of the rotating field, analogous to the pump beam heading error. Regarding the probe beam heading error, we define the angle that the probe beam rotates away from the x-axis as $\alpha$, with the sensor rotating in the xz-plane., as shown in Fig. 1. The spin projection along the probe beam can be written as $P_{prob}(t) = P_x(t)\cos\alpha + P_z(t)\sin\alpha$. By inserting Eqs. (3) and (4), and analyzing the first harmonic phase shift that depends on $\alpha$, we can determine the phase shift caused by the probe beam heading error

$$\phi_{prob}(t) \approx -\frac{B_m}{B_z} \tan\alpha \cdot \sin\omega_m t. \qquad (9)$$

Compared to Eq. (5), this is equivalent to transverse magnetic field $b_x$. Converting it into a magnetic field unit, it corresponds to a fictitious magnetic field $-\omega_m/\gamma \cdot \tan\alpha$ along the x-direction, which we refer to as a probe beam heading error.

**Eddy current.** Another systematic error is induced by eddy currents. The altering rotating field can generate eddy currents on the electrically conductive aluminum or $\mu$-metal magnetic shield, which in turn generates a magnetic field. The magnitude of the eddy current

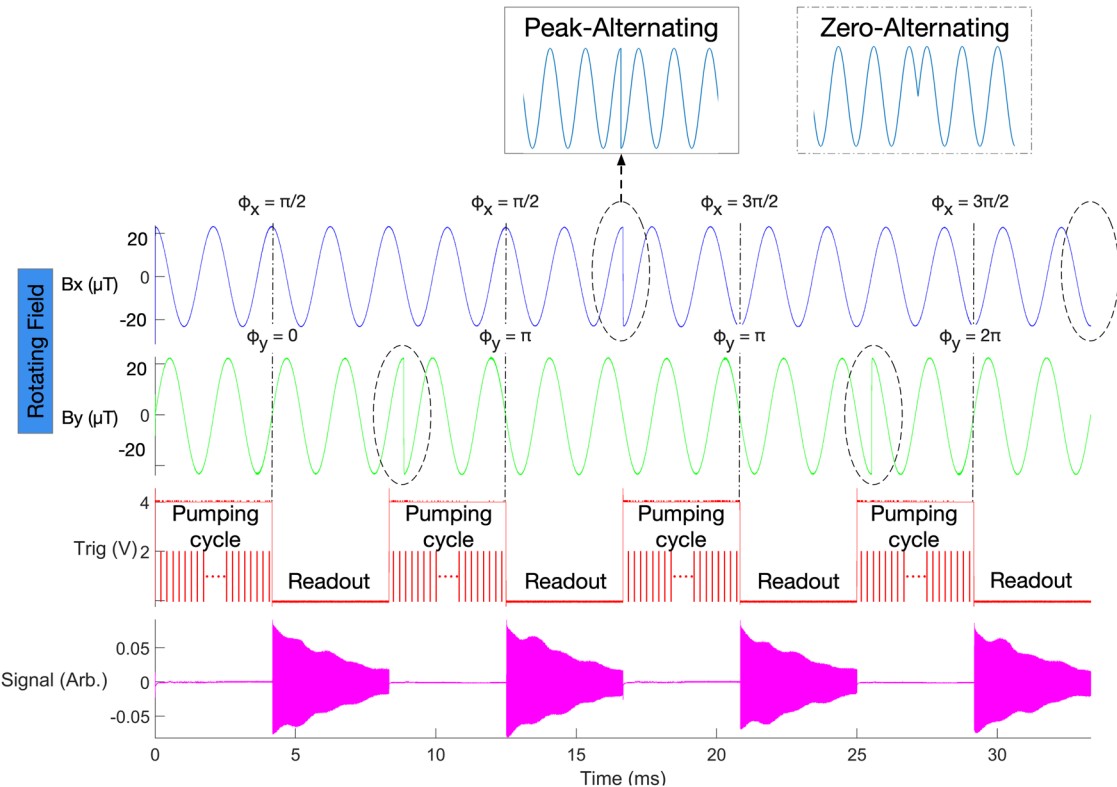

**Fig. 2 | Four-shot rotating field scheme.** Trig signal, when high after a rising edge, indicates the preparation phase: the AC heater heats the cell, and the pump laser polarizes the atoms. When low after a falling edge, the pump laser and AC heater are turned off, and the probe beam laser measures the FID signal. There are four shots with different start phases of $B_x$ and $B_y$. The start phases of the rotating field play a significant role as shown in Table 1; therefore, we indicate them with vertical dashed lines. This four-shot scheme is specifically developed to cancel out the systematic effects, such as Berry's phase shift, dynamic heading error, probe beam heading error, eddy current, and the systematic caused by the threshold voltage of the MDA. The magnetic field only alters during the peak values ("Peak-Altering") to suppress the eddy current caused by the rotating field. The signals during the preparation time are blanked out.

magnetic field highly depends on how the rotating magnetic field is altered. (More details in Sec. 2.3)

**Sign dependence of systematics.** We discovered that the signs of these systematic errors depend on the phases of the rotating field, while their absolute magnitudes remain unchanged. This characteristic offers an opportunity to eliminate these systematics. The sign of the $B_{PD}$ part of dynamic heading error depends on the rotating field's phases, but the sign of the static heading error does not. We introduce a four-shot scheme (shown in Fig. 2) by altering the phases of the rotating field in the order listed in Table 1. By averaging these four shots, the systematic errors can be canceled out.

**Experimental results**

To study systematics, a single-pass alkali vapor cell is placed at the center of a magnetic shield, which is a cubic cell with internal dimensions of $5 \times 5 \times 5$ mm$^3$. The cell contains a droplet of $^{87}$Rb and 688 Torr N$_2$ as quenching and buffer gas, respectively. The magnetic shield comprises a two-layer mu-metal shield and an innermost-layer aluminum shield designed to attenuate DC and AC magnetic fields from the environment.

The alkali cell is pumped by a sequence of pulses from a grating-stabilized diode laser, with a transverse relaxation time measured at $T_2 \approx 3$ ms. Inside the magnetic shield, a set of three coils provides a rotating magnetic field and a leading magnetic field. The cell is heated to 100 °C by an electric heater driven by AC at a frequency of 131.5 kHz. The AC heater is turned on during the pump time and turned off during the measurement time to reduce magnetic noise from the heater.

As shown in Fig. 1, a rotating magnetic field with an amplitude of ~18 μT and a freuqency of $\omega_m = 2\pi \times 480$ Hz is applied in the transverse plane, and a leading magnetic field $B_z$ is applied along the longitudinal direction. The total amplitude of the applied magnetic field is maintained at ~50 μT. The frequency of the pump beam pulse is set to 348 kHz, synchronized with the Larmor frequency to achieve the highest pumping rate. The duty cycle of a pump beam pulse is ~1.4%.

After the pump phase, the pump beam is switched off, and the free induction decay (FID) signal of the spin precession is measured using a linearly polarized probe beam. The probe beam, with a cross-sectional area of ~4 mm$^2$ and a power of around 2 mW, is generated by a vertical cavity surface-emitting laser (VCSEL). The optical rotation of the linearly polarized probe beam is measured by a balanced polarimeter consisting of a Wollaston prism and a quadrant photodiode. The signal from the photodiode is then passed through a differential low-noise amplifier and a high-pass filter with a cutoff frequency of 150 kHz.

The phase shift of the acquired signal relative to the Larmor frequency is analyzed using the HP 53310A modulation domain analyzer (MDA), which detects the zero-crossings of the precession signal and a reference signal of $\omega_{ref}$ that is close to the precession frequency. The phase shifts between the zero crossings of the precession signal and the reference signal are then calculated. The phase shift measurement of one block (four shots) of data is shown in Fig. 3.

The fitting model further refines the measured phase shifts, $\delta\Phi(t)$, for each shot from the MDA, modeling them as a function of measurement time to obtain: offset, slope ($\Phi_s$, Supplementary Table 2 in the Supplemental Information details the components of the slope), amplitudes of the first harmonics ($b_x \propto \Phi_x$ and $b_y \propto \Phi_y$), second harmonics of $\omega_m$, and amplitudes of the first harmonics of the hyperfine

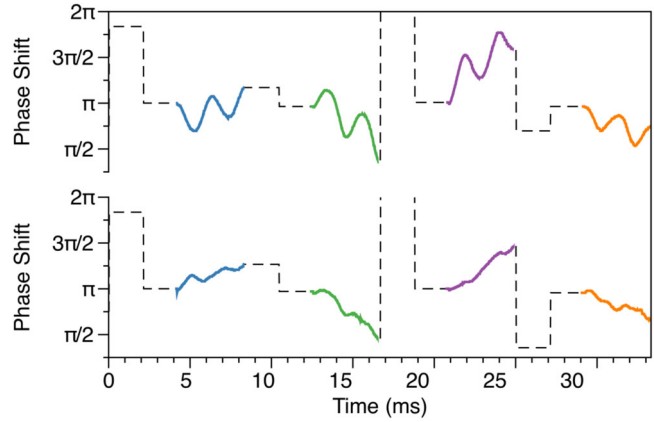

**Fig. 3 | Phase shift analysis.** Upper: when a magnetic field of -87 nT is applied in the y-direction, the phase shifts of the FID signal acquired in Fig. 2 are analyzed by the MDA as a function of time. Lower: when the magnetic fields in the x and y directions are roughly compensated. The first harmonic signals are suppressed and the second harmonic signal can be observed.

**Table 2 | The measurement results and predictions are based on rotating matrix simulations and analytical models**

| Systematic | Measured | Predicted |
|---|---|---|
| Berry's phase | 5.1 nT | 4.7 nT |
| Amplitude of $\sin 2\omega_m t$ | 35 mrad | 35 mrad |
| Static heading error $B_{SH}$ ($\beta \approx 24°$) | 2.7 nT | 2.9 nT |
| $|B_{PD}|$ ($\theta \approx 21°$, $\beta \approx 0$, $B_m$ initiates from Y) | 2.6 nT | 2.8 nT |

frequency $\omega_{hp}$ (the difference in precession frequency for the F = 1 and F = 2 atoms induces an exponentially decaying sinusoidal phase shift at a frequency of $\omega_{hp} = 2\pi \times 1.39$ kHz, see Supplementary Note 1 in the Supplemental Information).

$$\delta\Phi(t) = \Phi_{off} + \Phi_s \cdot t + \Phi_x \sin \omega_m t + \Phi_y \cos \omega_m t$$
$$+ \Phi_{2s} \sin 2\omega_m t + \Phi_{2c} \cos 2\omega_m t \qquad (10)$$
$$+ \Phi_{hps} \sin \omega_{hp} t + \Phi_{hpc} \cos \omega_{hp} t$$

The second harmonic of $\omega_m$ is also included in the fitting model because either the second harmonic phase shift described in Eq. (7) or the asymmetries of the rotating field in amplitude or phase can lead to a signal at a frequency of $2\omega_m$.

**Results of the systematics.** The experimental results for the systematics are presented in Table 2. The Berry phase will result in a phase shift that adds to the total magnetic field measurement. When $\theta \approx 21°$, magnetic field measurement systematic caused by Berry's phase shift is measured to be 5.1 nT, while Eq. (6) predicts 4.7 nT ($\phi_B/\gamma$), as shown in Table. 2.

The amplitude of the second harmonic of $\omega_m$ in phase shift is 35 mrad, which matches Eq. (7) prediction well.

The static component of the dynamic heading error is measured when the magnetometer sensor rotates by an angle $\beta = 24°$ in the yz-plane, resulting in a heading error of $B_{SH} = \mathcal{B}_{\mathcal{H}} \cos \theta \sin \beta$. From Eq. (8), the static heading error is estimated to be 2.9 nT, with an experimental result of 2.7 nT. The slightly lower experimental value is attributed to incomplete spin polarization. The predicted value for $|B_{PD}|$ is based on approximate analytical model detailed in Supplementary Table 3.

A rotating magnetic field can induce a rotating-field-phase-dependent heading error, $B_{PD}$. This effect is characterized in Table 3 with both experimental results and density matrix simulations[32]. The

**Table 3 | Measured dynamic heading error, primarily influenced by $B_{PD}$, along the total magnetic field with $\beta \approx 0°$, $\theta \approx 21°$**

| | Shot No. | $B_m$ initiates from Y | | $B_m$ initiates from X ($B_{tot} \perp S$) | |
|---|---|---|---|---|---|
| | | Exp (nT) | DM (nT) | Exp (nT) | DM (nT) |
| $B_{tot}$ | 1 | 2.6 | 3.0 | 0.1 | 0.0 |
| | 2 | 2.6 | 3.0 | 0.4 | 0.0 |
| | 3 | −2.6 | −3.0 | −0.1 | 0.0 |
| | 4 | −2.6 | −3.0 | −0.4 | 0.0 |
| Avg | | **0.0** | **0.0** | **0.0** | **0.0** |
| $B_x$ | 1 | −1.5 | −0.1 | −0.8 | 0.0 |
| | 2 | 0.4 | 0.1 | 0.8 | 0.0 |
| | 3 | −0.3 | −0.1 | −1.0 | 0.0 |
| | 4 | 1.6 | 0.0 | 1.3 | 0.0 |
| Avg | | **0.0** | **0.0** | **0.1** | **0.0** |
| $B_y$ | 1 | −0.1 | 0.1 | −0.1 | 0.0 |
| | 2 | 0.5 | 0.1 | 0.1 | 0.0 |
| | 3 | 0.1 | −0.1 | 0.1 | 0.0 |
| | 4 | −0.2 | −0.1 | 0.2 | 0.0 |
| Avg | | **0.1** | **0.1** | **0.1** | **0.0** |

Here the static part of dynamic heading error $B_{SH} \approx 0$. "Exp" is the experimental data, "DM" is the density matrix simulation result.

density matrix model, which represents an ensemble of spins in a mixed state, is described in detail in Supplementary Note 8 and Supplementary Tables of the supplemental Information.

When $1/T_2 < \omega_m \ll \omega_0$, the dynamic heading error can be interpreted as the spin precession plane adiabatically following the rotation of the total magnetic field $\mathbf{B_{tot}}$. This keeps the relative angle between $\mathbf{B_{tot}}$ and the spin precession plane constant, resulting in a steady precession frequency. Consequently, the dynamic heading error depends on the initial angle between the spin polarization and $\mathbf{B_{tot}}$, which remains equal to its initial value. We define $\Theta$ as the angle by which $\mathbf{B_{tot}}$ deviates from the position where $\mathbf{S}$ is perpendicular to $\mathbf{B_{tot}}$, as shown in Fig. 4a.

For example, when $\mathbf{B_m}$ begins along the x-axis ($\phi_x = \pi/2$, $\phi_y = 0$), the spins are polarized in a direction tilted by an angle $\beta$ away from the negative y-axis in the yz-plane, and $\mathbf{B_{tot}}$ initially lies in the xz-plane. Due to symmetry, whether $\mathbf{B_m}$ starts from the positive or negative x-axis, the relative angles between $\mathbf{S}$ and $\mathbf{B_{tot}}$ remain the same. In this configuration, the dynamic heading error equals the static heading error, $\mathcal{B}_{\mathcal{H}} \cos \theta \sin \beta$, with $B_{PD} = 0$. Under these conditions, the spin precession plane follows the rotation of the total magnetic field while keeping $\Theta$ constant. Specifically, if $\beta = 0$ initially, $\mathbf{S}$ stays perpendicular to $\mathbf{B_{tot}}$ throughout, resulting in $B_{DH} = 0$.

However, if $\mathbf{B_m}$ originates from the negative y-axis and $\mathbf{B_{tot}}$ is initially in the yz-plane, then $\mathbf{S}$ deviates from perpendicularity with $\mathbf{B_{tot}}$ by an angle $\Theta = \beta + \theta$. This results in a dynamic heading error of $\mathcal{B}_{\mathcal{H}} \sin(\beta + \theta)$, with $B_{SH} = \mathcal{B}_{\mathcal{H}} \cos \theta \sin \beta$, and $B_{PD} = \mathcal{B}_{\mathcal{H}} \cos \beta \sin \theta$. Conversely, if the rotating field begins from the positive y-axis, $\Theta = \beta - \theta$, with $B_{SH}$ remaining the same, but $B_{PD} = -\mathcal{B}_{\mathcal{H}} \cos \beta \sin \theta$. Table 1 provides a full overview of the sign dependence of the dynamic heading error.

The experimental results for the dynamic heading error are smaller than the analytical model and density matrix simulation because the actual spin polarization is smaller than the simulation, which assumes full polarization. For more data with different angles $\beta$ between the pump beam and the magnetic field, please refer to Supplementary Tables of the Supplemental Information.

We further evaluated the dynamic heading error effect in the transverse directions. A summary of the experimental results and the theoretical predictions related to $\mathbf{B_{tot}}$, based on the density matrix, is

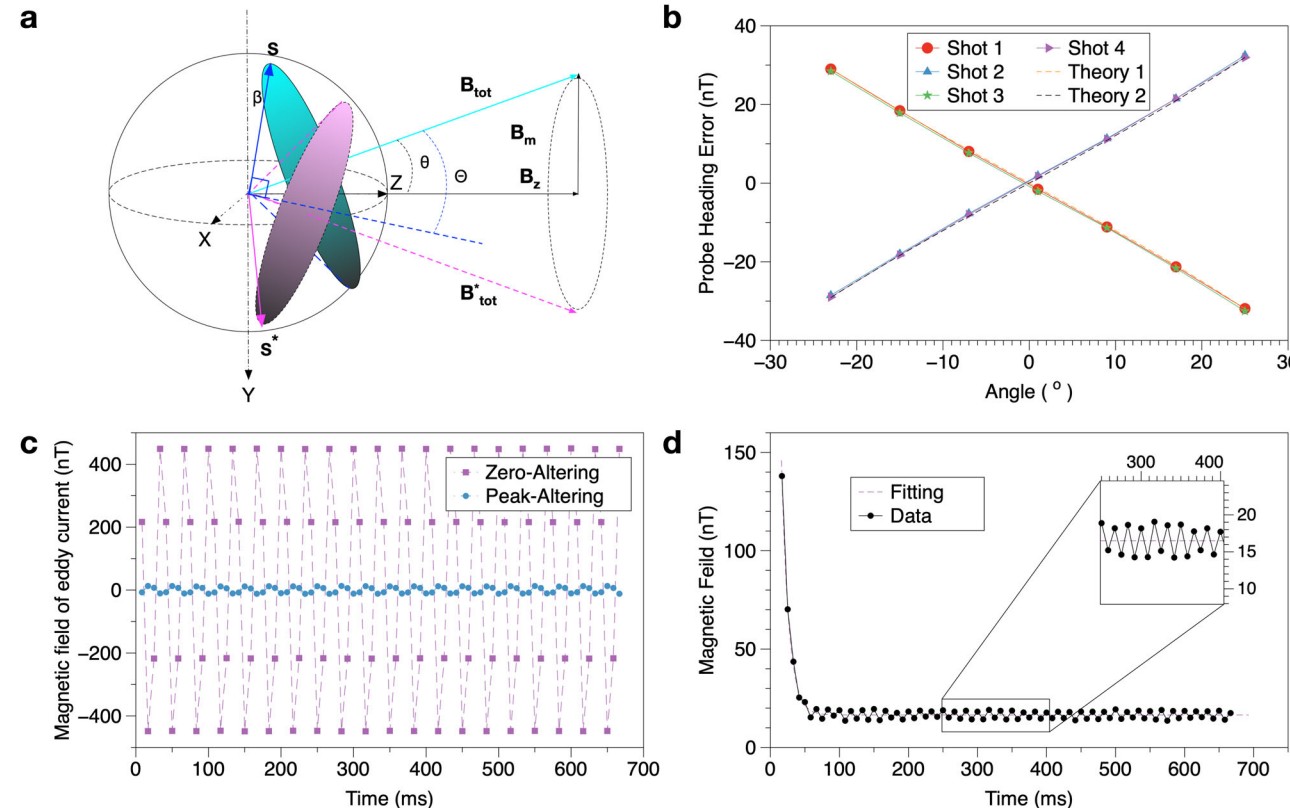

**Fig. 4 | Plots of systematic measurements. a** Dynamic heading error: The spin precession plane adiabatically follows $\mathbf{B_{tot}}$. **b** Measurement of the probe beam heading error: The solid points represent experimental results, while the dashed lines correspond to theoretical predictions based on the proposed model. **c** Eddy currents generated by different rotating field alterations: Measurement of the transverse magnetic field (y-axis) induced by eddy currents due to zero-altering and peak-altering modulations. **d** Eddy Current Time Constant: Measurement of the eddy current time constant during a single panorama measurement of MDA.

provided in Table 3. The dynamic heading error effect remains close to zero in $B_y$, irrespective of whether the rotating field begins parallel or perpendicular to $\mathbf{S}$. These minor magnetic field variations may result from systematic mismatches in the MDA's threshold voltage, leading to curvature that affects the fitting of the slope $\Phi_s$.

The variation in $B_x$ measurement from the density matrix simulation in each shot is due to the probe beam heading error ($\phi_{\mathrm{prob}}/\gamma$). For example, if the sensor is misaligned with an angle of $\alpha = 1°$, this results in an ~1.2 nT probe beam heading error in the x-direction. Additional data in Supplementary Tables of the Supplemental Information indicate that these magnetic field offsets maintain their signs even when the polarization direction is reversed, confirming that they are not caused by dynamic heading error. These systematic errors, however, can be eliminated through 4-shot averaging.

Overall, the rotating magnetic field can induce dynamic heading errors in total magnetic field measurements, consisting of a static heading error component and a rotating-field-phase-dependent component. The rotating-field-phase-dependent heading error, $B_{PD}$, can be eliminated either by selecting appropriate starting phases for the rotating field—ensuring the field initially aligns perpendicularly or maintains a constant angle relative to the spins with alternating start directions—or by averaging the results of four shots, as shown in Table 1. Importantly, the dynamic heading error does not affect measurement of the transverse magnetic fields, regardless of the starting phases of the rotating field.

To investigate the probe beam heading error, the sensor rotates about the y-axis within the xz-plane at an angle of $\alpha$. This error can introduce a systematic effect in transverse magnetic field measurements that depends on the angle $\alpha$, with its sign determined by the rotation direction of the rotating field. The measured probe heading error at $\omega_m = 2\pi \times 480$ Hz is shown in Fig. 4b, with the theoretical basis provided in Eq. (9). "Theory 1" corresponds to anti-clockwise rotations (Shots 1 and 3 in Table 1), given by $-\omega_m/\gamma \cdot \tan\alpha$, while "Theory 2" represents clockwise rotations (Shots 2 and 4 in Table 1), given by $\omega_m/\gamma \cdot \tan\alpha$. The probe beam heading error can be effectively canceled by averaging the measurements from two opposite rotating field directions.

The alternating fast rotating magnetic field can generate an eddy current in the electrically conductive magnetic shield. In our four-shot scheme, which alternates the rotation direction, there are two switches within four shots, as shown in Fig. 2. We found that the way the rotating magnetic field is altered affects the eddy current and its induced magnetic field. When the direction of the rotating field changes at the peak values of the field—from negative maximum to positive maximum and vice versa—we refer to this as "Peak-Altering" (Fig. 2, top left). In this case, the average magnetic field caused by the eddy current equals zero. However, if the direction change occurs when the rotating magnetic field is at zero, termed "Zero-Altering" (Fig. 2, top right), it leads to a significant magnetic field caused by the eddy current.

In our setup, the y-direction switch of $B_y$ occurs during the preparation times of the second and fourth shots, as shown in Fig. 2. In the "Zero-Altering" case, the second and fourth shots experience the maximum eddy current, while the first and third shots have smaller eddy currents, as they capture the decaying eddy current generated during the fourth and second shots. The sign of the eddy current-induced magnetic field depends on the rotation direction of the rotating field. A panorama measurement (80 shots) result from the MDA, shown in Fig. 4c, indicates that Zero-Altering can cause an eddy current magnetic field of ~450 nT. In contrast, Peak-Altering effectively suppresses the systematic effect of the eddy current. Any remaining

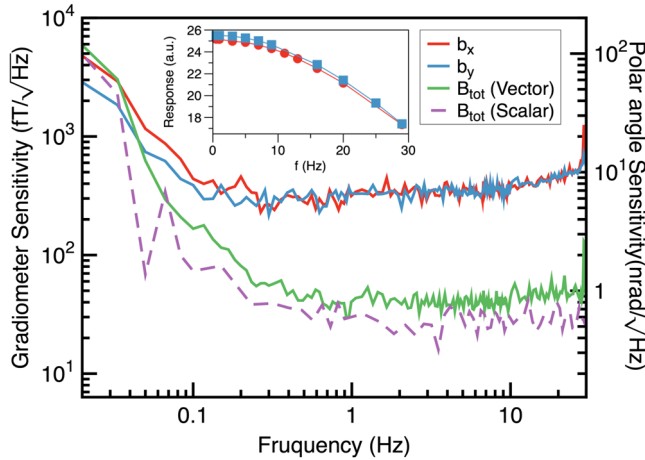

**Fig. 5 | Magnetic gradiometer and polar angle sensitivities.** The solid lines illustrate the sensitivities of the FRF vector magnetometer for the total magnetic field and the two transverse magnetic fields. The dashed line depicts the sensitivity of the magnetometer in scalar mode. The inset highlights the frequency response for $b_x$ and $b_y$. The polar angle sensitivities, applicable only to $b_x$ and $b_y$, are shown on the right-side y-axis and are calculated by dividing $b_x$ and $b_y$ by the total static magnetic field, $B_0$.

small eddy current magnetic field in Peak-Altering is due to imperfections, as instruments have a finite rise time during abrupt signal transitions.

To characterize the time constant of the eddy current, we applied a continuous rotating magnetic field with a constant phase (no alteration) over one panorama measurement. In Fig. 4d, 79 data points represent 79 individual shot measurements, with the first shot discarded due to phase shift overload in the MDA. Each shot measurement lasts 1/120 s. The 60 Hz line frequency produces periodic variations and harmonics, which are effectively averaged out over one panorama. Our setup's measured eddy current time constant is ~10.4 ms, corresponding to a cut-off frequency of 15 Hz.

To mitigate the eddy current's systematic effect, we designed a waveform shown in Fig. 2 to ensure that rotation direction alterations consistently occur at the peak values of the rotating magnetic field (Peak-Altering). By employing Peak-Altering and 4-shot averaging, we effectively reduce the eddy current's systematic effect.

**Results of magnetic field sensitivities.** A magnetic field gradiometer measures variations in magnetic field strength over a short distance, rather than the absolute field strength at a single location. This approach is highly effective for detecting local magnetic changes while minimizing the influence of uniform, distant magnetic interference[39]. To demonstrate the gradiometric measurement capability of the FRF vector magnetometer, we configured a setup using a multipass $^{87}$Rb cell and a high-power QPC laser with a 25 W output. The laser operates in pulsed mode to polarize the spins along the z-axis, enhancing spin alignment, and is turned off during the readout phase. After the pumping stage, $\pi/2$ magnetic field pulses are applied to tip the spins from the z-axis to the y-axis. Notably, for synchronized optical pumping that polarizes spins approximately perpendicular to the external magnetic field, the average power requirement is typically only in the tens of milliwatts, while achieving comparable performance.

Data acquisition is performed with two Carmel NK732 cards in place of the MDA, recording the zero-crossing data—including both frequency and phase information—from two magnetometers over a one-minute interval. We derive the gradiometer signal by subtracting the zero-crossing data from the two magnetometers, forming a first-order gradiometer that cancels common-mode magnetic field noise.

Assuming that any remaining noise is uncorrelated, we further divide this residual noise by $\sqrt{2}$ to assess the intrinsic magnetic field sensitivity of each channel[39].

The gradiometer sensitivities are shown in Fig. 5, with a sensitivity of 35 fT/$\sqrt{\text{Hz}}$ for the total field and 280 fT/$\sqrt{\text{Hz}}$ for the $B_x$ and $B_y$ components, respectively. Their noise spectrums are flat down to 1 Hz and 0.1 Hz, respectively. For comparison, a scalar sensitivity of 28 fT/$\sqrt{\text{Hz}}$ was achieved without the application of a rotating field.

## Discussion

In summary, we present a FRF vector geomagnetic magnetometer. This system is achieved by applying a rotating field to a scalar atomic magnetometer, enabling it to fully determine the magnetic field vector. While the modulation slightly degrades the scalar performance from 28 fT/$\sqrt{\text{Hz}}$ to 35 fT/$\sqrt{\text{Hz}}$, it provides two additional polar angles with resolutions of 6 nrad/$\sqrt{\text{Hz}}$. This enhancement allows the vector magnetometer to precisely measure two transverse polar angles, eliminating the static heading error common in scalar atomic magnetometers. The unique vector axes are defined by the rotation plane of the applied field, increasing stability compared to relying on mechanical coil orthogonality.

We provide a comprehensive study of the systematics and develop peak-altering fast rotating field modulation to cancel out these effects. Benefits from frequency measurement, such vector magnetometers can achieve high fractional resolution. Additionally, the vector magnetometer retains the accuracy and metrological advantages of scalar atomic magnetometers, such as inherent calibration.

Furthermore, the fundamental sensitivity of the vector magnetometer in total field measurement is identical to that of scalar magnetometers. Compared with approaches using sequential modulation, our proposed vector magnetometer is faster (higher bandwidth) and experiences fewer systematic effects. The FRF vector magnetometer can be further improved to reach quantum-limited sensitivity[40].

## Methods
### Shot noise limit
To derive the shot noise limit for the FRF magnetometer, we utilize the CRLB[41] along with photon shot noise considerations specific to balanced polarimeter detection[42]. Based on these principles, the power spectral density $\rho(\omega)$ achieves its minimum when the measurement duration $t \approx 2T_2$ (further details in Supplementary Note 4 of the Supplemental Information). For total magnetic field measurements, the sensitivity is expressed as

$$\delta B_{tot} \geq \frac{4}{\gamma k T_2 \sqrt{\Phi_{pr}}}, \tag{11}$$

where $k = l r_e c f n D(\nu)/2$, with $l$ being the probe beam path length through the cell, $r_e$ the classical electron radius, $f$ the typical oscillator strength of the D-line transition, and $D(\nu) = (\nu - \nu_0)/[(\nu - \nu_0)^2 + (\Delta\nu/2)^2]$, where $\Delta\nu$ is the optical full width at half maximum (FWHM) and $\nu_0$ is the D-line transition frequency. Here, $T_2$ is the transverse relaxation time, and $\Phi_{pr}$ denotes the photon flux per unit time. This noise limit also applies to scalar atomic magnetometers when analyzing the free induction decay (FID) signal.

When the measurement time matches the optimal duration for total magnetic field measurement ($t \approx 2T_2$), the sensitivity for transverse magnetic field measurements is

$$\delta B_{tran} \geq \frac{2\sqrt{2}\omega_m \csc\theta}{\gamma k \sqrt{\Phi_{pr}}}. \tag{12}$$

These derived sensitivities assume independent fitting of scalar and transverse magnetic fields. However, this assumption becomes

inaccurate if the modulation frequency is not sufficiently high. Analytical sensitivity solutions can be obtained by simultaneously fitting scalar and transverse magnetic fields using the Minimum-Variance Unbiased (MVU) estimator, as detailed in Supplementary Notes 4 and 5 of the Supplemental Information. The results show that when modulation frequency is sufficiently fast ($\omega_m \gtrsim \pi/T_2$), independent fitting yields comparable sensitivities.

Based on experimental parameters, we estimate the shot noise sensitivities of the FRF magnetometer to be $\delta B_{tot} = 0.04 \, \text{fT}/\sqrt{\text{Hz}}$ and $\delta B_{tran} = 0.8 \, \text{fT}/\sqrt{\text{Hz}}$. However, experimental measurements show sensitivities worse than these theoretical limits, primarily due to magnetic noise from current sources.

For instance, an SRS DC205 supplies -650 mA to generate the primary magnetic field along the z-axis, simulating a 50 $\mu$T Earth field. Despite modifications to reduce current noise, it remains a primary noise source, limiting overall sensitivity.

To compare, the fundamental sensitivity of the transverse magnetic field for sequential modulation (SM) vector magnetometers is given by (refer to Supplementary Note 6 of the Supplementary Information for details):

$$\delta B_{tran\_SM} = \frac{4\sqrt{2}}{\gamma k T_2 \sin\theta \sqrt{\Phi_{pr}}}. \tag{13}$$

If $\omega_m = \pi/T_2$, the sensitivity for transverse magnetic fields in the FRF vector magnetometer can be expressed as

$$\delta B_{tran\_FRF} = \frac{2\sqrt{2}\pi}{\gamma k T_2 \sin\theta \sqrt{\Phi_{pr}}}. \tag{14}$$

Thus, the FRF vector magnetometer achieves transverse field sensitivity comparable to that of sequential modulation vector magnetometers while offering a higher bandwidth and reduced susceptibility to systematic effects.

## Data availability
All data necessary to evaluate the conclusions of this study are included in the article, with supplementary data provided in tables and additional information included in the Supplementary Information. Any further details related to this study can be requested from the corresponding authors.

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

## Acknowledgements

The authors thank Dmitry Budker and Conlon Lorcan Oneill for their valuable comments. All authors acknowledge support from the Defense Advanced Research Projects Agency (DARPA) under contract No. HR001120C009 (M.R., T.K.). T.W. acknowledges support from A*STAR Career Development Fund (222D800028)(T.W.), Italy-Singapore science and technology collaboration grant (R23IOIR042)(T.W.), Delta-Q (C230917004, Quantum Sensing)(T.W.), and Competitive Research Program (NRF-CRP30-2023-0002)(T.W.).

## Author contributions

T.W. prepared the experimental setup and conducted tests for the FRF vector magnetometers, proposed and implemented the modulation scheme, modeled spin evolution using rotation matrix and density matrix simulations, investigated systematics, developed a model to explain the dynamic heading error, derived fundamental limits, analyzed the results, and drafted the manuscript. M.R. supervised the setup and experiments, contributed to spin evolution modeling using rotation matrices, and collaborated on discussions of modulation and dynamic heading error. W.L. assisted with modeling the static heading error and analyzing atomic polarization. M.L., T.K., and E.F. provided scalar magnetometer sensors, offered valuable experimental insights, and gave constructive feedback on manuscript revisions. All authors reviewed and approved the final manuscript.

## Competing interests

The authors declare no competing interests.
