## [Transparent Peer Review file · Nature Communications]

Pulsed Vector Atomic Magnetometer Using an Alternating Fast-Rotating Field

Corresponding Author: Dr Tao Wang

Version 0:

Reviewer comments:

Reviewer #1

(Remarks to the Author)

The authors report a vector atomic magnetometer based on free-induction-decay (FID) with high fractional sensitivity, probably the highest so far according to the table in the paper. By applying a fast-rotating magnetic field to a scalar atomic magnetometer, they achieved a sensitivity of $50 \text{ fT/Hz}^{1/2}$ for the total magnetic field. Simultaneously, resolutions of $8 \text{ nrad/Hz}^{1/2}$ for the two polar angles are obtained. This is very impressive performance achieved with an intelligent method. The main innovation here is to convert the measurements of both the field amplitude and direction into the measurement of a frequency. Since vector magnetometry is now becoming a topic of great interests, and how to obtain similar levels of high sensitivity with the scalar ones (without involving an overly complicated setup) remains to be challenging, the excellent results reported in the paper has the potential of being published in Nature Communications.

However, at the current stage, this manuscript is poorly written and requires huge effort to comprehend even for experts in atomic magnetometry. It is very difficult for a nonexpert to understand the physics picture behind the main idea of this study, because lots of important information is missing, such as the definition or description of technical terminologies, a general picture of the operating principle without referring to equations, and how are the measurements conducted etc., as I list below. The paper needs significant rewriting in order to be suitable for Nature Communications, otherwise it should be published in a more specialized journal.

1. In the title and introduction, the authors emphasize that this magnetometer has a high dynamic range. However, the manuscript does not provide any evidence (data) of the advantages of this vector magnetometer over other vector magnetometers in terms of dynamic range.

2. The measured total magnetic field is a composite magnetic field, which is a combination of the RF magnetic field B_m applied and the main magnetic field containing B_z (I do not understand why the transverse magnetic fields b_x and b_y are not included) according to Fig.1(a). Usually, the to-be-measured field of practical use is the one excluding the RF field, for example, in the application of Earth's field measurement. This means that both the direction and the amplitude of the B_m field needs to be known with high accuracy. The author should clarify which total magnetic field they are measuring.

3. In section 2.1, the derivation of Equation 4 is overly simplified. Readers can hardly understand the process of spin evolution. The important derivation process and key assumptions need to be presented somewhere such as in the supplementary material.

4. The authors emphasize that they have achieved the best polar angle sensitivity to date, as shown in Table 1. However, the manuscript does not provide any data to support this claim. The authors should clarify how is the angular sensitivity derived from the experiment data. Also, the sensitivity of the transverse magnetic field is quite low $400 \text{ fT/Hz}^{1/2}$, why?

5. Again in 2.3.2, vector magnetic gradiometer results are presented, but it is unclear how the data are extracted from single-channel results. Is it by subtracting the measured Larmor precession frequencies and phases? Also, in Fig.3. the bandwidth of the magnetometer seems to be much lower than results from similar setups in the same group. This should be explained. Does the applied rotating field affect this bandwidth? Also, with the current FID method, how can one measure a time-changing magnetic field?

6.Many terminologies need to be defined or briefly described (with proper references if necessary) at their first mention, such as heading error, how does Berry phase enter the problem, ω_m etc. The authors should do a thorough check on this issue.

7.Some key experimental parameters are missing, such as the temperature of the vapor cell, laser beam size and laser power for all lasers used, etc. The details of the experiment should be provided to an extent that it could be reproduced by others.

8.It seems that the scheme proposed in this paper are only limited to the case where the direction of the to-be-measured field B is already known to a good extent, because the applied RF field needs to be roughly in the plane perpendicular to B. So, does the fractional sensitivity depend on the direction of B if the RF field is fixed to a particular plane (without prior knowledge of the field direction)? The authors should discuss this issue.

9.The authors analyze the fundamental limits for the sensitivity of this vector free-spin-precession magnetometer. The actual value of this fundamental sensitivity using the experimental parameters/conditions should be provided. Meanwhile, the authors should further give a comparison of theoretical and experimental values, and discuss what are the main experimental reasons that keep the current results away from the fundamental limit. For example, how does the non-ideal polarization of the atoms affect the result?

10.Some minor issues: the quality of Fig.1 needs to be improved; there are quite some inconsistency in the format of the references.

Reviewer #2

(Remarks to the Author)

High Dynamic Range Vector Atomic Magnetometer with 1 part-per-billion Resolution in Earth Field Range

The Authors present a novel methodology for recoding data with an atomic magnetometer. Atomic spins are pumped along and orthogonal axis (y-axis) to a static field (z-axis), and the measurable signal comes from recording the free-induction-decay of these spins periodically when the pump is off. The application of a rotating field ω_m in a plane orthogonal to the static field enables measurement of vectorial field information (B_y and B_x) through careful analysis of the signals phase. Generally, the magnetometer is in the so-called 'bell-bloom' configuration and is used to measure dc magnetic fields. However, there is significant novelty in the complex measurement and signal processing used, resulting in a high-performance sensor. Most of the manuscript is dedicated to detailing the systematic effects on the measured magnetic field and listing their theoretical description in the supplementary material.

Overall the manuscript is dense, favouring brevity over clarity. This stylistic choice would make it challenging for a general reader; however, my issue is that the level of detail would also make it challenging to replicate the measurement.

We believe more can be done to make this easier to read, however scientifically it describes very important work and is a valuable addition to the research on atomic sensing, promising useful functionalise and performance. We recommend this for publication after the comments below are addressed and would encourage more detail on experimental processes, particularly around signal readout and data analysis.

Listed below are the few small technical comments that would improve the manuscript.

Q1) The value for ω_m is never given. It is understood that this value is set by T_2 , which is also not given.

Q2) '(the difference in precession frequency for the $F=1$ and $F=2$ atoms induces an exponentially decaying sinusoidal phase shift at a frequency of $\omega_{hp} = 2\pi \times 1.39$ kHz, see supplemental material Sec. I).' I understand the text here and the accompanying in Sup. Mat. Sec. 1 describes the beatnote seen in the FFT due to atoms precessing in the $F=1$ and $F=2$ hyperfine groundstates. Is there not additional frequencies due to the non-linear Zeeman effect experienced at $wL = 348$ kHz within each F state?

Q3) Is there a reason the inner shield layer is aluminium, not an outer layer? The sensor is affected by eddy currents which would be less in μ -metal, or ideally ferrite.

Q4) What temperature is the cell?

Q5) Heating is applied at 131.5kHz. Is there a specific reason for this frequency and would heating frequencies much greater than Larmor frequencies (e.g. 1MHz) result in less noise, e.g. harmonics.

Q6) figure 1 (b) has units of G while μT is used in the text.

Q7) Figure 1 (b): Generally this figure is complicated.

a) What are the dot-dashed vertical lines representing?

b) The red square wave line, assumed to be the 'Trig (V)' signal isn't described. The B_x cosine switch lines up with trigger pulse (~ 17 ms), but the B_y cosine switch does not (~ 8 ms and 25ms).

c) The 'Laser/Readout' pulse diagram could be confused with the pump beam pulse (at 348kHz), rather than describing 'Laser' as many pumping cycles.

4) In the signal, there is clearly some beating in the signal. What is this, the signal from ω_m .

Q8) AWGN is defined after (pg2) first use on pg 1 in Sup. Mat.

Q9) Figure 2 (c) line plots are very faint when printed.

Q10) How is the cosine-switch implemented?

Q11) Is the bandwidth of the sensor limited to ~ 30 Hz due to the ~ 35 ms four-shot measurement? It would be fair to include upper bandwidth limit Table 1.

Q12) How stable does the target magnetic field need to be during the four-shot measurement. Changes in the field will accrue phase shifts between the shots.

Version 1:

Reviewer comments:

Reviewer #1

(Remarks to the Author)

I thank the authors for considering all my questions and made corresponding changes in the paper. I recommend publication of the paper in Nature Communications, since the vector magnetometer reported here provides so far the best fractional sensitivity. As such, my only remaining suggestion is to put table 1 (that shows the comparison with other vector magnetometers) back in the paper, but in the supplementary material due to space limit of the main text.

Reviewer #2

(Remarks to the Author)

We thank the authors for addressing the comments. More technical details have added to overall give better description of the experiment. One point is that the reported 25W optical power required is hugely demanding from a SWaPC perspective, limiting the practicality of this method. This point should be highlighted in the manuscript.

The paper is dense and requires expertise of the field to understand the text, however we believe the manuscript is valuable and suitable for publication.

Authors' Response to Reviews of

Pulsed Vector Atomic Magnetometer Using an Alternating Fast-Rotating Field

Tao Wang, Wonjae Lee, Mark Limes, Tom Kornack, Elizabeth Foley, Michael Romalis
Nature Communications,

RC: *Reviewers' Comment*, AR: Authors' Response, □ Manuscript Text

1. Reviewer #1

The authors report a vector atomic magnetometer based on free-induction-decay (FID) with high fractional sensitivity, probably the highest so far according to the table in the paper. By applying a fast-rotating magnetic field to a scalar atomic magnetometer, they achieved a sensitivity of $50 \text{ fT}/Hz^{1/2}$ for the total magnetic field. Simultaneously, resolutions of $8 \text{ nrad}/Hz^{1/2}$ for the two polar angles are obtained. This is very impressive performance achieved with an intelligent method. The main innovation here is to convert the measurements of both the field amplitude and direction into the measurement of a frequency. Since vector magnetometry is now becoming a topic of great interests, and how to obtain similar levels of high sensitivity with the scalar ones (without involving an overly complicated setup) remains to be challenging, the excellent results reported in the paper has the potential of being published in *Nature Communications*.

However, at the current stage, this manuscript is poorly written and requires huge effort to comprehend even for experts in atomic magnetometry. It is very difficult for a nonexpert to understand the physics picture behind the main idea of this study, because lots of important information is missing, such as the definition or description of technical terminologies, a general picture of the operating principle without referring to equations, and how are the measurements conducted etc., as I list below. The paper needs significant rewriting in order to be suitable for *Nature Communications*, otherwise it should be published in a more specialized journal.

AR: Thank you for your valuable feedback and for recognizing the significance of our work. We agree that the manuscript requires more clarity, and we will revise it by improving the explanations of key concepts, providing clearer definitions, and offering a more intuitive description of the experimental setup. We appreciate your suggestions and will work to ensure the paper is more accessible to a wider audience. We also added discription of operation principal:

The intuitive classical model underlying the operating principle of the FRF vector magnetometer is illustrated in Fig. 1a. The scalar magnitude of the total magnetic field, $|\mathbf{B}_{\text{tot}}|$, depends on the angle between the rotating field \mathbf{B}_m and the residual magnetic field \mathbf{B}_0 . When a small residual transverse magnetic field $\Delta\mathbf{B}$ —comprising b_x and b_y —is present, $|\mathbf{B}_{\text{tot}}|$ varies with oscillating components at the rotation frequency of \mathbf{B}_m . The phases and amplitudes of these oscillations are influenced by the angle between \mathbf{B}_0 and \mathbf{B}_m . Consequently, the Larmor precession frequency, proportional to $|\mathbf{B}_{\text{tot}}|$, also displays oscillating components: a residual field in the x-direction, b_x , produces an in-phase oscillation, while a residual field in the y-direction induces an out-of-phase component. This model provides a fundamental description of the magnetometer's operation. However, for precise measurements, several systematic effects—some even resulting from hyperfine interactions—require

in-depth analysis to account for and minimize potential measurement biases.

RC: *(1) In the title and introduction, the authors emphasize that this magnetometer has a high dynamic range. However, the manuscript does not provide any evidence (data) of the advantages of this vector magnetometer over other vector magnetometers in terms of dynamic range.*

AR: Thank you for the insightful feedback. The high dynamic range highlighted here is specifically intended as a comparison to ultrahigh sensitivity atomic magnetometers, particularly those in the spin-exchange relaxation-free (SERF) regime, where the dynamic range is inherently limited. This limitation stems from the requirement for a high spin-exchange rate relative to the Larmor precession frequency, which restricts their applicability. By contrast, our pulsed vector magnetometer directly measures the precession frequency, theoretically enabling a much higher dynamic range and enhanced fractional resolution. In response to the reviewer's suggestion, we will revise the title to remove the reference to high dynamic range. Additionally, we will clarify in the manuscript that our vector magnetometer was successfully tested within the Earth's magnetic field strength range.

RC: *(2) The measured total magnetic field is a composite magnetic field, which is a combination of the RF magnetic field B_m applied and the main magnetic field containing B_z (I do not understand why the transverse magnetic fields b_x and b_y are not included) according to Fig.1(a). Usually, the to-be-measured field of practical use is the one excluding the RF field, for example, in the application of Earth's field measurement. This means that both the direction and the amplitude of the B_m field needs to be known with high accuracy. The author should clarify which total magnetic field they are measuring.*

AR: Thank you for the insightful comment! The total magnetic field measured, B_{tot} , in Fig. 1(a) indeed includes the transverse components b_x and b_y , and we have updated Fig. 1(a) to clarify this. The rotating field is generated by a high-accuracy current source, ensuring precise control of the B_m amplitude. The measured magnetic field can be determined by the polar angles of the total magnetic field relative to the rotation plane and the known amplitude of the rotating magnetic field. While the polar angles are precisely measurable by the FRF magnetometers, one could further improve alignment using a coil system to adjust the orientation of the rotating-field plane, maintaining it perpendicular to the external magnetic field by keeping the measured polar angle near zero. If any AC noise couples from the rotating field, such as slow amplitude drift, it will be visible in the sensitivity spectrum shown in Fig. 3. Conversely, an inaccuracy in the DC value of B_m would primarily introduce a systematic offset. Notably, the FRF vector magnetometer allows for easy switching between scalar and vector modes by enabling or disabling the rotating field, which facilitates self-calibration by alternating between modes and effectively reducing systematics introduced by amplitude inaccuracies in the rotating field. Experimental results (see Sec. II.2) confirm that AC noise from rotating field modulation minimally impacts sensitivity, with total magnetic field measurement sensitivity only slightly decreasing from $28 \text{ fT}/\sqrt{\text{Hz}}$ to $35 \text{ fT}/\sqrt{\text{Hz}}$.

RC: *(3) In section 2.1, the derivation of Equation 4 is overly simplified. Readers can hardly understand the process of spin evolution. The important derivation process and key assumptions need to be presented somewhere such as in the supplementary material.*

AR: Thank you for the reviewer's comment, we have provided a more detailed derivation of Equation 4 in the supplemental material Section II.

We assume the spins are fully polarized along the negative y-axis. The magnetic field initially points along the z-axis, while a rotating field originates from the x-axis and rotates counterclockwise in the xy-plane. We consider the behavior of polarization in a magnetic field, where it undergoes precession in

response to the applied field. This magnetic field is composed of both a static and a rotating component. In the presence of these components, the polarization vector evolves according to specific rotations. The spin evolution, without considering spin relaxations, can be expressed as:

$$\mathbf{P}(t) = \mathcal{R}[\theta, \hat{z}] \cdot \mathcal{R}[\psi, \mathbf{B}_{\text{tot}}] \cdot (-\hat{y}), \quad (1)$$

where $\mathcal{R}[\phi, \mathbf{v}]$ is a 3D rotation matrix for an anti-clockwise rotation of ϕ degrees around the vector $\mathbf{v} = \{i, j, k\}$,

$$\mathcal{R}[\phi, \mathbf{v}] = \begin{pmatrix} \frac{i^2 + (j^2 + k^2) \cos(\phi)}{i^2 + j^2 + k^2} & \frac{-k \sin(\phi) \sqrt{i^2 + j^2 + k^2} - ij \cos(\phi) + ij}{i^2 + j^2 + k^2} & \frac{j \sin(\phi) \sqrt{i^2 + j^2 + k^2} - ik \cos(\phi) + ik}{i^2 + j^2 + k^2} \\ \frac{k \sin(\phi) \sqrt{i^2 + j^2 + k^2} - ij \cos(\phi) + ij}{i^2 + j^2 + k^2} & \frac{(i^2 + k^2) \cos(\phi) + j^2}{i^2 + j^2 + k^2} & \frac{-i \sin(\phi) \sqrt{i^2 + j^2 + k^2} - jk \cos(\phi) + jk}{i^2 + j^2 + k^2} \\ \frac{-j \sin(\phi) \sqrt{i^2 + j^2 + k^2} - ik \cos(\phi) + ik}{i^2 + j^2 + k^2} & \frac{i \sin(\phi) \sqrt{i^2 + j^2 + k^2} - jk \cos(\phi) + jk}{i^2 + j^2 + k^2} & \frac{(i^2 + j^2) \cos(\phi) + k^2}{i^2 + j^2 + k^2} \end{pmatrix} \quad (2)$$

For the applied rotating field in the x-y plane, we have $B_x = B_m \sin(\omega_m t + \phi_x)$, $B_y = B_m \sin(\omega_m t + \phi_y)$, and $\text{Abs}[\phi_x - \phi_y] = \pi/2$. Therefore, $\mathbf{B}_{\text{tot}} = (B_m, 0, B_z - \omega_m/\gamma)$. Here, γ represents the gyromagnetic ratio, and \hat{x} , \hat{y} and \hat{z} are the unit vectors along the x, y and z axes, respectively. We define $\theta = \omega_m t$ and $\psi = \gamma t \sqrt{B_m^2 + (B_z - \omega_m/\gamma)^2}$. Eventually, the spin projections can be written as:

$$P_x(t) = \hat{x} \cdot \mathbf{P}(t) = \cos \omega_0 t \sin \omega_m t + \frac{\gamma B_z - \omega_m}{\omega_0} \sin \omega_0 t \cos \omega_m t, \quad (3)$$

$$P_z(t) = \hat{z} \cdot \mathbf{P}(t) = -\frac{\gamma B_m}{\omega_0} \sin \omega_0 t, \quad (4)$$

where $\omega_0 = \gamma |\mathbf{B}_{\text{tot}}|$. We have now included the first-order solution for the additional rotation caused by the transverse fields b_x and b_y . This solution is derived based on the integration of the rotation angle, providing a more precise account of the transverse field effects in the measurement.

$$\begin{aligned} \psi &\approx \gamma t \sqrt{B_m^2 + b_x^2 + b_y^2 + (B_z - \omega_m/\gamma)^2} - \gamma t \frac{B_m b_y \cos \omega_m t}{\sqrt{B_m^2 + B_z^2}} \\ &+ \gamma t \frac{B_m b_x \sin \omega_m t}{\sqrt{B_m^2 + B_z^2}} \end{aligned} \quad (5)$$

By inserting residual transverse magnetic fields b_x and b_y into Eq.1, we can get

$$\begin{aligned} P_x(t) &\approx \cos \left(\omega_0 t + \gamma \frac{B_m b_y \cos \omega_m t - B_m b_x \sin \omega_m t}{\omega_m \sqrt{B_m^2 + B_z^2}} \right) \\ &\sin \omega_m t + \sin \left(\omega_0 t + \gamma \frac{B_m b_y \cos \omega_m t - B_m b_x \sin \omega_m t}{\omega_m \sqrt{B_m^2 + B_z^2}} \right) \\ &\cos \omega_m t \cdot \frac{\gamma B_z - \omega_m}{\omega_0}, \end{aligned} \quad (6)$$

where $\omega_0 = \gamma |\mathbf{B}_{\text{tot}}| = \gamma \sqrt{B_m^2 + b_x^2 + b_y^2 + (B_z - \omega_m/\gamma)^2}$.

RC: (4) *The authors emphasize that they have achieved the best polar angle sensitivity to date, as shown in Table*

1. However, the manuscript does not provide any data to support this claim. The authors should clarify how is the angular sensitivity derived from the experiment data. Also, the sensitivity of the transverse magnetic field is quite low $400 \text{ fT}/\text{Hz}^{1/2}$, why?

AR: We demonstrated a sensitivity of $280 \text{ fT}/\sqrt{\text{Hz}}$ (adjusted for uncorrelated differential measurements, where differential noise is reduced by a factor of $\sqrt{2}$), within the plane of the rotating field at a leading field strength of approximately $50 \mu\text{T}$. Any slight tilt of the leading field in the transverse direction can be detected at this sensitivity level, corresponding to polar angle sensitivities of $6 \text{ nrad}/\sqrt{\text{Hz}}$. As noted above, one can adjust the rotating field plane to be perpendicular to the external magnetic field. Any minor tilt, causing a change in the polar angle, will produce a small magnetic field projection onto the rotating field plane. This polar angle can be estimated by dividing the transverse magnetic field by the leading field. Importantly, it is not the absolute sensitivity that is paramount here but rather the outstanding fractional resolution in the transverse direction compared to the total field. Additionally, our approach offers a unique advantage: even in the presence of coil imperfections or non-orthogonality, as long as the coils remain linear, the rotating field maintains a distinct rotation plane, effectively defining the vector axes within our setup.

RC: **(5) Again in 2.3.2, vector magnetic gradiometer results are presented, but it is unclear how the data are extracted from single-channel results. Is it by subtracting the measured Larmor precession frequencies and phases? Also, in Fig.3. the bandwidth of the magnetometer seems to be much lower than results from similar setups in the same group. This should be explained. Does the applied rotating field affect this bandwidth? Also, with the current FID method, how can one measure a time-changing magnetic field?**

AR: We have added more detailed setup schematics in the Supplemental Material to help readers better understand the implementation of the gradiometer measurement. The gradiometer signal was extracted from the zero-crossing data (including frequency and phase information) of two magnetometers. The bandwidth is currently limited due to the four-shot averaging scheme; with a repetition rate of 120 Hz , four shots take $1/30 \text{ s}$, resulting in a sampling rate of 30 Hz . Consequently, the bandwidth for measuring time-varying magnetic fields is reduced. However, the bandwidth can be increased by raising the repetition rate and reducing the preparation/pumping time.

Data acquisition is performed with two Carmel NK732 cards in place of the MDA, recording the zero-crossing data—including both frequency and phase information—from two magnetometers over a one-minute interval. We derive the gradiometer signal by subtracting the zero-crossing data from the two magnetometers, forming a first-order gradiometer that cancels common-mode magnetic field noise. Assuming that any remaining noise is uncorrelated, we further divide this residual noise by $\sqrt{2}$ to assess the intrinsic magnetic field sensitivity of each channel.

RC: **(6) Many terminologies need to be defined or briefly described (with proper references if necessary) at their first mention, such as heading error, how does Berry phase enter the problem, ω_m etc. The authors should do a thorough check on this issue.**

AR: We added descriptions before each terminology. Berry's phase occurs in the context of electron spins when the system undergoes adiabatic, cyclic evolution. In this experiment, as the magnetic field direction traces a closed loop, rotating in a circle, the electron's spin state acquires a Berry's phase in addition to the dynamical phase. For instance:

[Berry's phase shift] The Berry phase shift refers to the geometric phase acquired by a spin system during an adiabatic, cyclic evolution [35]. As the magnetic field rotates slowly, the spins adiabatically

follow the field. At the end of a complete cycle, when the field returns to its original configuration, the spins acquire a phase shift. If the system undergoes cyclic adiabatic evolution at a constant frequency, the Berry phase accumulates linearly over time. Thus, the Berry phase term as a function of time, from Eq. 3, can be written as

[Heading errors] The heading error of atomic magnetometers refers to the dependence of the measured magnetic field values on the orientation of the sensor relative to the magnetic field [36]. The pump beam heading error is well-studied and is primarily caused by nonlinear Zeeman splitting and the difference between Zeeman resonances in the two hyperfine ground states. We set nuclear Landé factor $g_I \approx 0$ [37] and the heading error can be well described by $B_{SH} = B_H \sin \beta$, where

There's an analogous "heading error" effect from the probe beam. This error creates a systematic effect where the measured transverse magnetic field depends on the relative angle between the probe beam and the plane of the rotating field, analogous to the pump beam heading error. Regarding the *probe beam heading error*, we define the angle that the probe beam rotates away from the x-axis as α , with the sensor rotating in the xz-plane., as shown in Fig. ??(b). The spin projection along the probe beam can be written as $P_{prob}(t) = P_x(t) \cos \alpha + P_z(t) \sin \alpha$. By inserting Eqs. 3 and 4, and analyzing the first harmonic phase shift that depends on α , we can determine the phase shift caused by the probe beam heading error

A magnetic field gradiometer measures the variation in magnetic field strength over a short distance, rather than the absolute field strength at a single location. This method is effective for detecting local magnetic changes while reducing the influence of uniform, distant magnetic interference [39]. To demonstrate the gradiometric measurement capability of the FRF vector magnetometer, we configured a setup using a multipass ^{87}Rb cell and a high-power QPC laser with a 25 W output. The laser, operating in pulsed mode, polarizes the spins along the z-axis to enhance spin alignment and is then turned off during the readout phase. Following this pumping stage, $\pi/2$ magnetic field pulses are applied to tip the spins from the z-axis to the y-axis.

RC: (7) *Some key experimental parameters are missing, such as the temperature of the vapor cell, laser beam size and laser power for all lasers used, etc. The details of the experiment should be provided to an extent that it could be reproduced by others.*

AR: We added more details about the experimental setup. The cell temperature is 100°C , the power of the QPC laser is 25 W. The probe beam has a cross-sectional area of approximately 4 mm^2 and a power of approximately 2 mW.

The alkali cell is pumped by a sequence of pump pulses from a grating-stabilized diode laser. Inside the magnetic shield, a set of three coils provides a rotating magnetic field and a leading magnetic field. The cell is heated to 100°C by an electric heater driven by AC at a frequency of 131.5 kHz. The AC heater is turned on during the pump time and turned off during the measurement time to reduce magnetic noise from the heater.

After the pump phase, the pump beam is switched off, and the free induction decay (FID) signal of the spin precession is measured using a linearly polarized probe beam. The probe beam, with a cross-sectional area of approximately 4 mm^2 and a power of around 2 mW, is generated by a vertical cavity surface-emitting laser (VCSEL). The optical rotation of the linearly polarized probe beam is measured by a balanced polarimeter consisting of a Wollaston prism and a quadrant photodiode. The signal from the photodiode is then passed through a differential low-noise amplifier and a high-pass filter with a cutoff frequency of 150 kHz.

To demonstrate the magnetic gradiometer measurement of the FRF vector magnetometer, a multipass ^{87}Rb cell is used. A high-power QPC laser with a 25 W output operates in pulsed mode, polarizing the spins along the z-axis to enhance spin polarization; it is switched off during readout. Following the pump, $\pi/2$ magnetic field pulses are applied to tip the spins from the z-axis to the y-axis.

RC: (8) *It seems that the scheme proposed in this paper are only limited to the case where the direction of the to-be-measured field \mathbf{B} is already known to a good extent, because the applied RF field needs to be roughly in the plane perpendicular to \mathbf{B} . So, does the fractional sensitivity depend on the direction of \mathbf{B} if the RF field is fixed to a particular plane (without prior knowledge of the field direction)? The authors should discuss this issue.*

AR: The magnetometer can initially be aligned approximately with the external magnetic field. Any misalignment or deviation from perpendicularity will introduce a magnetic field component along the rotating field plane, appearing as transverse magnetic fields that the FRF vector magnetometer can measure. However, sensitivity to the total magnetic field is notably higher than that for transverse measurements. To address cases where the direction of the target field \mathbf{B} is not well known, a coil system generating a rotating field in any arbitrary direction can be used. This system would continuously align the rotation plane perpendicularly to the external magnetic field by actively measuring the transverse components, ensuring optimal alignment and sensitivity even when the precise direction of \mathbf{B} is initially unknown.

RC: (9) *The authors analyze the fundamental limits for the sensitivity of this vector free-spin-precession magnetometer. The actual value of this fundamental sensitivity using the experimental parameters/conditions should be provided. Meanwhile, the authors should further give a comparison of theoretical and experimental values, and discuss what are the main experimental reasons that keep the current results away from the fundamental limit. For example, how does the non-ideal polarization of the atoms affect the result?*

AR: We have added a discussion on this point. Nonideal polarization can result in incomplete suppression of spin-exchange relaxation, leading to a shorter T_2 time and, consequently, reduced fundamental sensitivity. However, in our setup, the magnetic noise from the current source driving the coils remains the dominant factor.

Based on experimental parameters, we estimate the shot noise sensitivities of the FRF magnetometer to be $\delta B_{tot} = 0.04 \text{ fT}/\sqrt{\text{Hz}}$ and $\delta B_{tran} = 0.8 \text{ fT}/\sqrt{\text{Hz}}$. However, experimental measurements show sensitivities worse than these theoretical limits, primarily due to magnetic noise from current sources.

For instance, an SRS DC205 supplies approximately 650 mA to generate the primary magnetic field along the z-axis, simulating a $50 \mu\text{T}$ Earth field. Despite modifications to reduce current noise, it remains a primary noise source, limiting overall sensitivity.

RC: (10) *Some minor issues: the quality of Fig.1 needs to be improved; there are quite some inconsistency in the format of the references.*

AR: We have enhanced the quality of the figures in Fig. 1 and reviewed the references to ensure consistent formatting.

2. Reviewer #2

The Authors present a novel methodology for recoding data with an atomic magnetometer. Atomic spins are pumped along and orthogonal axis (y-axis) to a static field (z-axis), and the measurable signal comes from recording the free-induction-decay of these spins periodically when the pump is off. The application of a rotating field ω_m in a plane orthogonal to the static field enables measurement of vectorial field information (B_y and B_x) through careful analysis of the signals phase. Generally, the magnetometer is in the so-called ‘bell-bloom’ configuration and is used to measure dc magnetic fields. However, there is significant novelty in the complex measurement and signal processing used, resulting in a high-performance sensor. Most of the manuscript is dedicated to detailing the systematic effects on the measured magnetic field and listing their theoretical description in the supplementary material. Overall the manuscript is dense, favouring brevity over clarity. This stylistic choice would make it challenging for a general reader; however, my issue is that the level of detail would also make is challenging to replicate the measurement. We believe more can be done to make this easier to read, however scientifically it describes very important work and is a valuable addition to the research on atomic sensing, promising useful functionalise and performance. We recommend this for publication after the comments below are addressed and would encourage more detail on experimental processes, particularly around signal readout and data analysis.

AR: Thank you for your thoughtful feedback and for recognizing the novelty and significance of our work. We appreciate your suggestion to provide more clarity and detail, particularly around the experimental processes, signal readout, and data analysis. We will revise the manuscript to improve readability and include additional information to ensure the measurement procedures are more transparent and easier to replicate.

Listed below are the few small technical comments that would improve the manuscript.

RC: (1) *The value for ω_m is never given. It is understood that this value is set by T_2 , which is also not given.*

AR: The value of $\omega_m = 2\pi \times 480$ Hz was mentioned in Sec II.C.1. T_2 was measured to be approximately 3 ms. We have now also added this information in Sec II.C.

The alkali cell is pumped by a sequence of pulses from a grating-stabilized diode laser, with a transverse relaxation time measured at $T_2 \approx 3$ ms. Inside the magnetic shield, a set of three coils provides a rotating magnetic field and a leading magnetic field.

As shown in Fig. 1a, a rotating magnetic field with an amplitude of approximately $18 \mu\text{T}$ and a frequency of $\omega_m = 2\pi \times 480$ Hz is applied in the transverse plane, while a leading magnetic field B_z is applied along the longitudinal direction.

RC: (2) *‘(the difference in precession frequency for the $F=1$ and $F=2$ atoms induces an exponentially decaying sinusoidal phase shift at a frequency of $\omega_{hp} = 2\pi \times 1.39$ kHz, see supplemental material Sec. I).’ I understand the text here and the accompanying in Sup. Mat. Sec. 1 describes the beatnote seen in the FFT*

due to atoms precessing in the $F=1$ and $F=2$ hyperfine groundstates. Is there not additional frequencies due to the non-linear Zeeman effect experienced at $\omega_L = 348\text{kHz}$ within each F state?

AR: The hyperfine phase shift is indeed prominent in our experimental data and aligns with observations reported in Phys. Rev. A 103, 063103. Additionally, the nonlinear Zeeman effect is present in our results, manifesting as the dynamic heading error that we propose and investigate in this manuscript. This effect contributes additional frequency components within each F state. This frequency arises from the nonlinear Zeeman shift and is distinct from the beat frequency $\omega_{hp} = 2\pi \times 1.39\text{kHz}$ caused by the precession difference between the $F=1$ and $F=2$ hyperfine ground states.

RC: *(3) Is there a reason the inner shield layer is aluminium, not an outer layer? The sensor is affected by eddy currents which would be less in mu-metal, or ideally ferrite.*

AR: Using a mu-metal magnetic shield for the innermost layer could indeed reduce Johnson magnetic noise and potentially minimize eddy currents due to its lower electrical conductivity. However, an aluminum innermost layer was the option available for this experiment. Additionally, by implementing the proposed peak-altering modulation along with shot-to-shot averaging, eddy current effects can be effectively suppressed. Moreover, the dominant noise sources in the current setup are primarily from magnetic noise generated by the current sources.

RC: *(4) What temperature is the cell?*

AR: The cell temperature is 100°C , we've added this to the revised manuscript.

The alkali cell is pumped by a sequence of pump pulses from a grating-stabilized diode laser. Inside the magnetic shield, a set of three coils provides a rotating magnetic field and a leading magnetic field. The cell is heated to 100°C by an electric heater driven by AC at a frequency of 131.5 kHz . The AC heater is turned on during the pump time and turned off during the measurement time to reduce magnetic noise from the heater.

RC: *(5) Heating is applied at 131.5kHz . Is there a specific reason for this frequency and would heating frequencies much greater than Larmor frequencies (e.g. 1MHz) result in less noise, e.g. harmonics.*

AR: Thank you for your comment. The magnetometer operates in pulsed mode, with the AC heater active only during the preparation phase and inactive during the readout phase, so the heating frequency does not affect the system. Shifting to a frequency of 1 MHz would decrease heating efficiency due to increased impedance. Additionally, by measuring the zero-crossings of the Larmor precession frequency, the signal remains unaffected by temperature fluctuations.

RC: *(6) figure 1 (b) has units of G while μT is used in the text.*

AR: Thank you for your comment. I have updated the unit in Figure 1(b) to ensure consistency with the manuscript's use of μT .

RC: *(7) Figure 1 (b): Generally this figure is complicated. a) What are the dot-dashed vertical lines representing? b) The red square wave line, assumed to be the 'Trig (V)' signal isn't described. The B_x cosine switch lines up with trigger pulse (17ms), but the B_y cosine switch does not (8ms and 25ms). c) The 'Laser/Readout' pulse diagram could be confused with the pump beam pulse (at 348kHz), rather than describing 'Laser' as many pumping cycles. d) In the signal, there is clearly some beating in the signal. What is this, the signal from ω_m .*

AR: a) The dashed lines represent the starting phases of the applied magnetic field in the x and y directions for each

shot after the pump pulse. Timing is crucial in this scheme, as the relationship between the starting phases of the rotating field and the signs of the systematics is detailed in Table I. By adjusting these starting phases and averaging over four shots, we can effectively cancel out these systematics. As discussed in the dynamic heading error section, the starting phase has a significant effect on the dynamic heading error. Selecting the optimal starting phase, where the initial spins are perpendicular to the total magnetic field, can eliminate the rotating-phase-dependent heading error, B_{PD} .

b) The “Trig” signal, shown as a red square wave, indicates the preparation phase when high after a rising edge: during this phase, the AC heater heats the cell, and the pump laser polarizes the atoms. When low after a falling edge, the pump laser and AC heater are turned off, and the probe laser measures the FID signal. The peak-altering is generally applied at the end of each shot measurement; while it doesn’t need to align precisely with the rising edge of the Trig signal, the falling edge sets the starting phase for the rotating field. With a measured eddy current time constant of 10.4 ms (see Sec II.C.1), the peak-altering remains necessary to prevent eddy currents from affecting subsequent measurements.

(a) Three coils determine the x, y, and z directions. The sensor head sits on a rotating stage with the cell positioned at the stage’s center. The sensor head can freely rotate in the xz and yz-planes. The angle between the sensor head’s central axis and the z-axis is denoted as α . β is the angle by which the pump beam moves away from the position where the pump laser is perpendicular to the magnetic field. θ represents angle between the axis of rotation and the rotating field vector. Notably, WP stands for Wollaston prism, and Q-PD stands for quadrant photodiode. (b) Trig signal, when high after a rising edge, indicates the preparation phase: the AC heater heats the cell, and the pump laser polarizes the atoms. When low after a falling edge, the pump laser and AC heater are turned off, and the probe beam laser measures the FID signal. There are four shots with different start phases of B_x and B_y . The start phases of the rotating field play a significant role; therefore, we indicate them with vertical dashed lines. This four-shot scheme is specifically developed to cancel out the systematic effects, such as Berry’s phase shift, dynamic heading error, probe beam heading error, eddy current, and the systematic caused by the threshold voltage of the MDA. The magnetic field only alters during the peak values (“Peak-Altering”) to suppress the eddy current caused by the rotating field. The signals during the preparation time are blanked out. (c) **Upper:** When a magnetic field of -87 nT is applied in the y-direction, the phase shifts of the FID signal acquired in b) are analyzed by the MDA as a function of time. **Lower:** when the magnetic fields in the x and y directions are roughly compensated. The first harmonic signals are suppressed and the second harmonic signal can be observed.

c) Thank you for the suggestion, we revised it to pumping cycle, and updated Fig. 1b.

d) The beating envelope at frequency ω_m is primarily induced by the applied rotating field. By using a high-pass filter, the signal at ω_m is removed, leaving the component near the Larmor precession frequency. Non-zero transverse magnetic fields introduce an oscillating component in the Larmor precession frequency, affecting the zero-crossings and resulting in periodic phase shifts, as shown in Fig. 1c.

RC: (8) *AWGN is defined after (pg2) first use on pg 1 in Sup. Mat.*

AR: Thank you for your comment. We have revised the supplemental document accordingly.

RC: (9) *Figure 2 (c) line plots are very faint when printed.*

AR: In response to the reviewer’s feedback, we have revised Figure 2(c) to improve readability, especially when printed.

RC: (10) *How is the cosine-switch implemented?*

- AR: The cosine-switch (which we have renamed to peak-altering) is implemented using Python code to generate a 24-bit, 192 kHz stereo WAV file, where the left and right channels correspond to B_x and B_y , respectively. This digital signal is sent through a USB audio bridge to a balanced MQA DAC and power amplifier. Each channel is then connected to a $1200\mu\text{F}$ capacitor to filter out the DC signal before being connected to the coils.
- RC: *(11) Is the bandwidth of the sensor limited to 30 Hz due to the 35 ms four-shot measurement? It would be fair to include upper bandwidth limit Table 1.*
- AR: Yes, the four-shot measurement time determines the duration needed to obtain a single data point, which directly impacts the bandwidth. The bandwidth depends on the repetition rate; thus, by increasing this rate and reducing the preparation time, the measurement bandwidth can be enhanced. We chose to remove Table 1 to reduce the length of an already extensive paper and to focus more on clearly presenting the measurement scheme of the FRF vector magnetometer and its systematics.
- RC: *(12) How stable does the target magnetic field need to be during the four-shot measurement. Changes in the field will accrue phase shifts between the shots.*
- AR: There is no specific stability requirement for the target magnetic field. After each shot, the spins relax and are then repolarized by the pump beam, with the phase shift analyzed after each pump cycle. This approach ensures that phase shifts do not accumulate between shots.

Authors' Response to Reviews of

Pulsed Vector Atomic Magnetometer Using an Alternating Fast-Rotating Field

Tao Wang, Wonjae Lee, Mark Limes, Tom Kornack, Elizabeth Foley, Michael Romalis
Nature Communications, MS# NCOMMS-24-26722A,

RC: *Reviewers' Comment*, AR: Authors' Response, Manuscript Text

1. Reviewer #1

I thank the authors for considering all my questions and made corresponding changes in the paper. I recommend publication of the paper in *Nature Communications*, since the vector magnetometer reported here provides so far the best fractional sensitivity. As such, my only remaining suggestion is to put table 1 (that shows the comparison with other vector magnetometers) back in the paper, but in the supplementary material due to space limit of the main text.

AR: Thank you for your kind feedback and for recommending our paper for publication in *Nature Communications*. As per your suggestion, we will include Table 1, which compares our results with other vector magnetometers, in the Supplementary Material to maintain conciseness in the main text. We greatly appreciate your insightful comments and support.

2. Reviewer #2

We thank the authors for addressing the comments. More technical details have added to overall give better description of the experiment. One point is that the reported 25W optical power required is hugely demanding from a SWaPC perspective, limiting the practicality of this method. This point should be highlighted in the manuscript.

The paper is dense and requires expertise of the field to understand the text, however we believe the manuscript is valuable and suitable for publication.

AR: Thank you for your thoughtful feedback and for recognizing the value of our manuscript. We appreciate your suggestion to highlight the high optical pumping power in the text.

The elevated optical power requirement in our experiment is due to the failure of the vacuum thermal isolation for an alternative scalar magnetometer. As a temporary measure, we employed a scalar sensor with the pump laser aligned in the same direction as the total magnetic field. This configuration was necessitated by the compact sensor head's long and narrow form factor, which made it difficult to install in the magnetic shield with the pump laser oriented roughly perpendicular to the external magnetic field while also keeping the cell centered within the shield. Consequently, a higher-power laser was required for effective pumping. After the pumping phase, a $\pi/2$ tipping pulse was applied to tip the spins into the transverse direction.

The scalar sensor originally intended for this experiment, prior to the vacuum failure, demonstrated identical performance to those used in <https://doi.org/10.1103/PhysRevApplied.14.011002> and in Lee, W. Ultra-high

Sensitivity Atomic Magnetometers (Princeton University, 2022). For context, the scalar magnetometer described in <https://doi.org/10.1103/PhysRevApplied.14.011002> is rated at approximately 5 W total power consumption, with the majority of power usage attributed to microcontrollers.

We will include this clarification in the revised manuscript to ensure transparency and provide context. Thank you again for your valuable feedback.